# Mitochondrial pyruvate carrier is required for optimal brown fat thermogenesis

Vanja Panic[1], Stephanie Pearson[1], James Banks[1], Trevor S Tippetts[2], Jesse N Velasco-Silva[1], Sanghoon Lee[1], Judith Simcox[1], Gisela Geoghegan[1], Claire L Bensard[1], Tyler van Ry[1], Will L Holland[2], Scott A Summers[2], James Cox[1], Gregory S Ducker[1], Jared Rutter[1,3], Claudio J Villanueva[1,4]*

[1]Department of Biochemistry, University of Utah, Salt Lake City, United States; [2]Department of Nutrition and Integrative Physiology, University of Utah, Salt Lake City, United States; [3]Howard Hughes Medical Institute, University of Utah, Salt Lake City, United States; [4]Department of Integrative Biology and Physiology, University of California, Los Angeles, Los Angeles, United States

**Abstract** Brown adipose tissue (BAT) is composed of thermogenic cells that convert chemical energy into heat to maintain a constant body temperature and counteract metabolic disease. The metabolic adaptations required for thermogenesis are not fully understood. Here, we explore how steady state levels of metabolic intermediates are altered in brown adipose tissue in response to cold exposure. Transcriptome and metabolome analysis revealed changes in pathways involved in amino acid, glucose, and TCA cycle metabolism. Using isotopic labeling experiments, we found that activated brown adipocytes increased labeling of pyruvate and TCA cycle intermediates from U$^{13}$C-glucose. Although glucose oxidation has been implicated as being essential for thermogenesis, its requirement for efficient thermogenesis has not been directly tested. We show that mitochondrial pyruvate uptake is essential for optimal thermogenesis, as conditional deletion of *Mpc1* in brown adipocytes leads to impaired cold adaptation. Isotopic labeling experiments using U$^{13}$C-glucose showed that loss of MPC1 led to impaired labeling of TCA cycle intermediates. Loss of MPC1 in BAT increased 3-hydroxybutyrate levels in blood and BAT in response to the cold, suggesting that ketogenesis provides an alternative fuel source to compensate. Collectively, these studies highlight that complete glucose oxidation is essential for optimal brown fat thermogenesis.

*For correspondence: cvillanueva@ucla.edu

Competing interests: The authors declare that no competing interests exist.

## Introduction

The ability to thermoregulate has allowed mammals to thrive in cold regions of the world. Brown adipose tissue (BAT) thermogenesis is an energy demanding process that has been key to the evolution and survival of mammals (*Gaudry et al., 2019*; *Oelkrug et al., 2015*; *Barnett and Dickson, 1989*). With the excess calorie intake associated with a western diet, mechanisms that promote energy expenditure in the cold will provide attractive therapeutic interventions to treating metabolic diseases associated with obesity (*Cypess et al., 2009*; *Vijgen et al., 2011*). Cold exposure triggers the activation of the sympathetic nervous system to secrete norepinephrine, which signals through the β3-adrenergic receptor (β3-AR) and stimulates production of cyclic AMP (cAMP) (*Townsend and Tseng, 2014*; *Londos et al., 1985*). cAMP promotes the activation of protein kinase A (PKA), which in turn upregulates transcription of thermogenic pathways and leads to the activation of lipolysis (*Cannon and Nedergaard, 2004*; *Zhang et al., 2005*). Free fatty acids released can directly activate Uncoupling Protein 1 (UCP1), which uncouples the electron transport chain to generate heat (*Fedorenko et al., 2012*; *Klaus et al., 1991*; *Busiello et al., 2015*). Cold exposure stimulates uptake of both glucose, triglyceride(TG)-rich lipoproteins and free fatty acids from the blood (*Labbé et al., 2015*; *Heine et al., 2018*; *Ferré et al., 1986*). While the relative contribution and importance of FFA

as a BAT fuel source has been extensively studied (*Bartelt et al., 2011*; *Khedoe et al., 2015*; *Townsend and Tseng, 2014*; *Lee et al., 2015*), our understanding of metabolic fate of glucose and the importance of its catabolism in thermogenesis in vivo remains unknown.

Human brown fat was only believed to be found in newborns, but now we appreciate that adults have brown adipose tissue, a discovery that was made through use of glucose tracer ($^{18}$F-fluoro-deoxyglucose) and positron-emission tomographic and computed tomographic (PET–CT) scans (*Cypess et al., 2009*; *Virtanen et al., 2009*). In addition, it was previously recognized that cold exposure could lower blood glucose levels in adults (*Martineau and Jacobs, 1989*). The role of glucose uptake on metabolism has been explored in vitro using immortalized brown adipocytes where siRNAs targeting *Slc2a1* (GLUT1), *Slc2a4* (GLUT4), *Hk2* (hexokinase2), or *Pkm* (pyruvate kinase, muscle) (enzymes catalyzing the first and the last step of glycolysis) demonstrated the importance of glycolysis, as β3-AR agonist failed to increase glucose uptake and oxygen consumption in these cells (*Winther et al., 2018*). However, there is no adequate in vivo model demonstrating the importance of BAT glycolysis or glucose oxidation on adaptive thermogenesis. We will address this question in vivo by blocking pyruvate import into mitochondria of brown adipocytes by knocking out the mitochondrial pyruvate carrier (MPC).

MPC is a multimeric complex in the inner mitochondrial membrane that consists of MPC1 and MPC2 subunits (*Bricker et al., 2012*; *Herzig et al., 2012*; *Schell et al., 2014*). Deletion of either subunit leads to instability of a functional MPC complex. MPC links the end product of glycolysis to glucose oxidation by transporting pyruvate into the mitochondrial matrix (*Mowbray, 1975*). Loss-of-function studies targeting MPC1 or MPC2 has been shown to limit mitochondrial pyruvate transport in yeast, flies and mammals (*Herzig et al., 2012*; *Bricker et al., 2012*). Once in the mitochondria, pyruvate is decarboxylated to acetyl-CoA for further processing in the TCA cycle to generate NADH and fuel ATP production by OXPHOS complexes. Alternatively, cytosolic pyruvate can be reduced to lactate by lactate dehydrogenase complex A (LDHA), a process commonly upregulated in cancer cells (*Vander Heiden et al., 2009*). While it is clear that cold exposure or direct stimulation of β3-AR stimulates glucose utilization by BAT in both humans (*Cypess et al., 2009*; *Saito et al., 2009*) and rodents (*Mirbolooki et al., 2014*; *Vallerand et al., 1990*), it is not clear how important glucose oxidation is during thermogenesis nor what the metabolic fate of glucose is in activated BAT. Recently, comparative metabolomics analysis has shown that activation of BAT led to increased levels of the TCA cycle intermediate succinate; however, it is unclear whether glucose-derived TCA cycle intermediates are required for thermogenesis (*Mills et al., 2018*).

In this study, we use comprehensive metabolomics analysis of BAT and serum from mice housed at different temperatures, to gain insight into the metabolic pathways altered with cold exposure. We find changes in glucose, amino acid, and TCA cycle intermediates in BAT. Using [U-$^{13}$C]-glucose, we found increased glycolytic and TCA cycle metabolism during BAT stimulation. To test whether glucose oxidation is required for thermogenesis, we generated mice lacking mitochondrial pyruvate carrier one subunit (MPC1) in brown adipose tissue. We found that mice lacking MPC1 in BAT are cold sensitive, indicating that pyruvate import into the mitochondria is essential for efficient thermogenesis. Furthermore, when we profiled serum and BAT metabolites of MPC1-null mice, we found elevated 3-hydroxybutyrate levels. Prior studies supporting a role for ketogenesis in thermogenesis, suggests an alternative carbon source that compensates for the loss of pyruvate transport. Together this study provides new insights into the metabolic fate of glucose in brown adipose tissue during activation of thermogenesis in response to acute cold exposure.

## Results

### Cold-induced changes in transcriptome and metabolite profiling of BAT

To systematically profile the transcriptional changes that are altered in response to acute cold exposure, we measured steady state levels of RNA in BAT from mice at room temperature (24℃) or cold (4℃) for 5 hr. We found that 1907 transcripts were upregulated with cold exposure, while 3273 were decreased (*Supplementary file 1* and *supplementary file 2*). Hierarchical clustering and Principal Component Analysis (PCA) revealed that the gene expression patterns in cold room and room temperature exposed BATs form two distinctive and independent clusters (*Figure 1—figure supplement 1A and B*). Using Gene Set Enrichment Analysis (GSEA), we found that cold exposure

stimulated distinct transcriptional changes in BAT that involve various aspects of metabolism. Notable changes include induction of glucose metabolic process, sphingolipid metabolism, amino acid metabolism, and cellular respiration, while pathways involved in cell cycle control, DNA repair, and glycoprotein metabolism were downregulated (*Figure 1A and B*).

To test whether steady state levels of metabolic intermediates were altered, we used targeted GC-MS analysis to complete comprehensive metabolic profiling of BAT (*Figure 1C and D*) and serum (*Figure 1E and F*) from mice across different temperatures (30°C, 23°C, and 4°C). The BAT metabolome showed elevated levels of glycolytic intermediates, TCA cycle intermediates, ketone bodies, and branched chain amino acids when mice were challenged with the cold (*Figure 1C*).

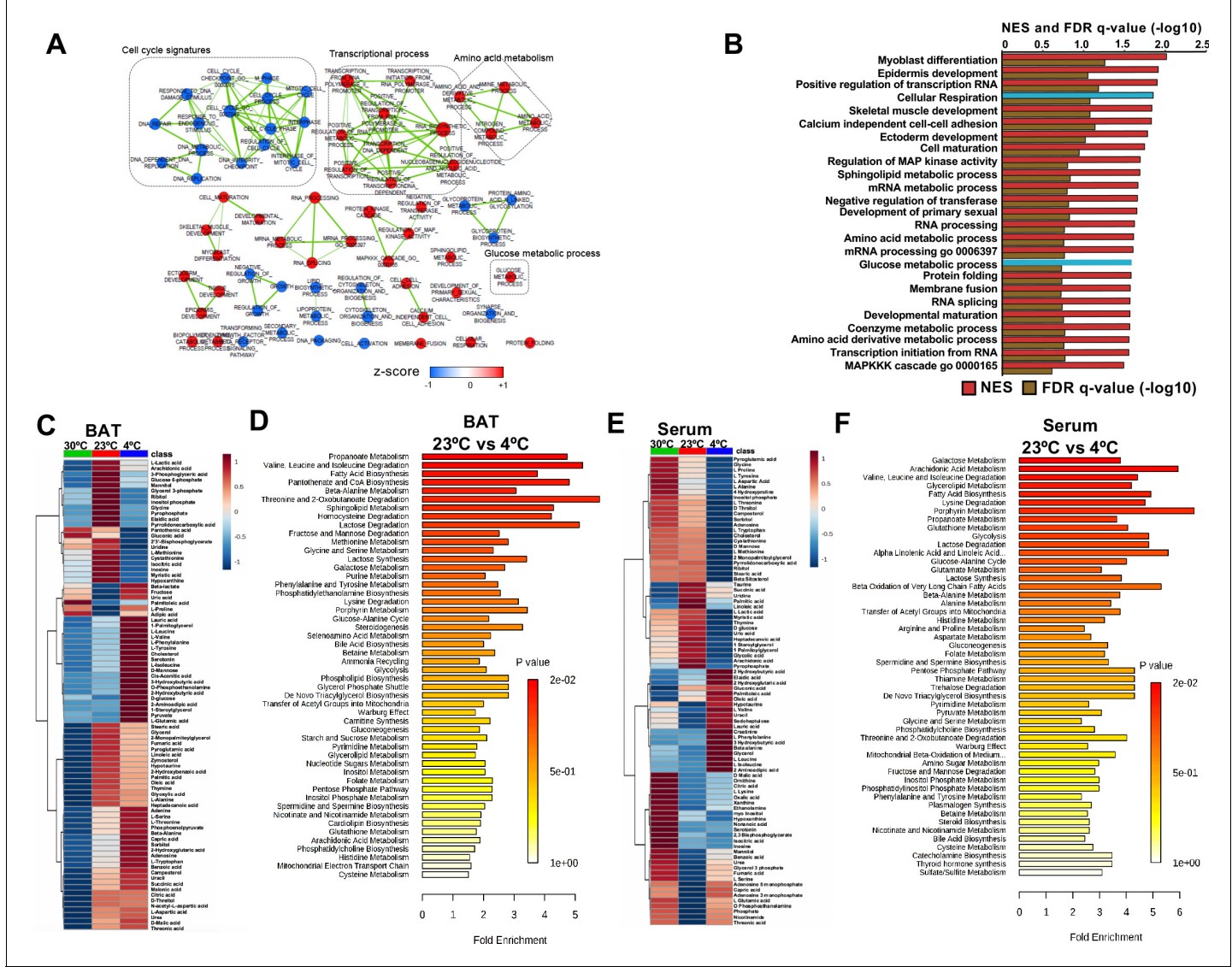

**Figure 1.** Transcriptome and metabolomics analysis of brown fat shows increased carbohydrate metabolism and glycolytic metabolism during cold exposure. (a) Network visualization of enriched biological pathways altered with cold exposure in BAT (N = 5). (b) GSEA pathway analysis of differentially expressing genes (FDR < 0.05) in BAT at 4°C versus room temperature (N = 5). (c) Heat map of relative normalized changes in BAT (c) and serum (e) metabolites at 30°C, 23°C, and 4°C. Dendograms illustrate hierarchical clustering of pattern similarity across metabolites (left) and conditions (top). Each column represents average within the group (N = 5 per group). Data was sum normalized, log transformed, and autoscaled. (d) MSEA pathway analysis of metabolites in BAT (d) and serum (f) from mice at 4°C versus room temperature (N = 5).

The online version of this article includes the following figure supplement(s) for figure 1:

**Figure supplement 1.** RNRNA-seq analysis of brown adipose tissue from mice at room temperature or with cold exposure.

Notably, amino acids like tyrosine, alanine, threonine, and tryptophan were increased in BAT, while their levels decreased in serum with cold exposure. Perhaps, BAT uptake could lead to their depletion in the blood. Similar to a recent report (*Yoneshiro et al., 2019*), we observed that branched chain amino acids, including Valine, Leucine, and Isoleucine were elevated in BAT, while only Leucine and Valine were upregulated in serum (*Figure 1C and E*). To identify metabolic pathways that changed with cold exposure, we used Metabolite Set Enrichment Analysis (MSEA) to compare metabolomes of BAT and serum from mice at room temperature (24°C) and cold (4°C) (*Figure 1D and F*). Both BAT and serum were enriched for pathways involved in amino acid, fatty acid, nucleotide, and glucose metabolism. Notably, glucose and pyruvate levels in BAT were elevated in response to 4°C, while both glucose and pyruvate levels were similar between mice housed at 30°C and 23°C. This finding would suggest that there is an increase in the rate of pyruvate synthesis in response to the cold (*Figure 1C*). A list of measured metabolites from BAT and serum are detailed in *supplementary file 3* and *supplementary file 4*.

The observed transcriptional and metabolite changes point to a reliance on pathways involved in carbohydrate metabolism (*Figure 2A*). This prompted further analysis of glucose catabolism in brown adipocytes under aerobic conditions in response to a β3-AR agonist CL-316,243 (*Figure 2B*). In vitro tracing experiments using [U-$^{13}$C]-Glucose showed that activation of brown adipocytes treated with CL-316,243 had significant $^{13}$C-glucose-derived M+3 isotopologues of $^{13}$C-Pyruvate, $^{13}$C-Lactate and $^{13}$C-Glycerol-3-Phosphate. Differentiated brown adipocytes that were treated with CL-316,243 had more than 50% of pyruvate and lactate labeled. Surprisingly, there was little alanine labeling from [U-$^{13}$C]-glucose, despite the rise in M+3 $^{13}$C-Alanine in response to β3-AR activation (*Figure 2B*). During incubation with [U-$^{13}$C]-Glucose, there was depletion of M+6 glucose in the media after CL-316,243 administration, while M+3 pyruvate in the media increased, but did not

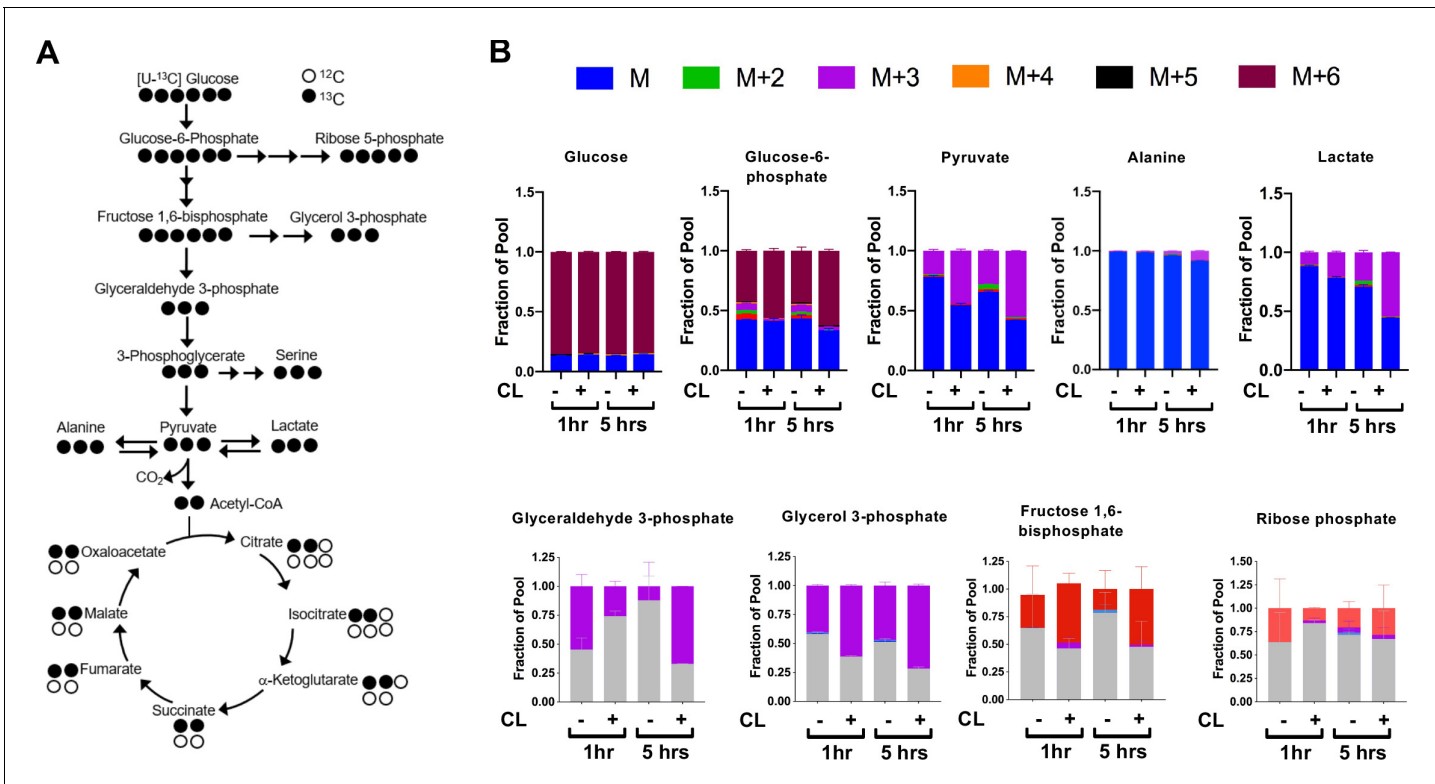

**Figure 2.** CL-316,243 stimulation of brown adipocytes leads to increased $^{13}$C-glucose flux. (a) Atom mapping for [U-$^{13}$C] glucose tracing into glycolysis and the TCA cycle. White balls are $^{12}$C atoms. Black balls are $^{13}$C atoms. (b) Tracing analysis from U-$^{13}$C glucose in differentiated brown adipocytes treated with vehicle or 100 nM CL-316,243 for 5 hours(N = 3).

The online version of this article includes the following figure supplement(s) for figure 2:

**Figure supplement 1.** Metabolite tracing of brown adipocytes in response to β3-Adrenergic Receptor agonist.

respond to CL-316,243 treatment (*Figure 2—figure supplement 1A*). To test whether M+3 lactate derived from [U-$^{13}$C]-Glucose was being released into the media, we measured media M+3 lactate, and found that CL-316,243 increased the release of M+3 lactate into the media when compared to vehicle (*Figure 2—figure supplement 1B*). These results suggest that activation of thermogenesis in brown adipocytes leads to increased lactate synthesis and secretion.

To address whether conditions that increase oxidative metabolism correlate with MPC levels, we measured the expression of *Mpc1* and *Mpc2* in BAT of C57BL6 mice challenged with thermoneutrality (30°C) or cold exposure (4°C) for 1 day or 1 week. Using real-time PCR, we found that both *Mpc1* and *Mpc2* expression had increased in BAT (*Figure 3A*). This was accompanied by induction of thermogenic transcripts, including *Ucp1* and *Dio2*, while *Cidea* expression was unchanged (*Figure 3A*). In contrast, thermoneutrality (30°C) decreased *Mpc2* and *Ucp1* expression, while *Mpc1* was unchanged (*Figure 3B*). Similarly, we saw increased protein expression of MPC1, MPC2, and UCP1 in BAT after 1 week of cold exposure (*Figure 3C* and *Figure 2—figure supplement 1A*). In contrast, another mitochondrial protein, Cytochrome C, remained unchanged after a similar cold exposure. The increased expression of MPC1 may provide additional pyruvate transport and oxidative capacity for sustaining prolonged thermogenesis in BAT.

## BAT-selective deletion of *Mpc1* leads to cold sensitivity and impaired glucose handling

To test whether MPC is required for thermogenesis, we generated mice with conditional deletion of *Mpc1* in BAT by crossing *Mpc1*$^{F/F}$ mice (*Gray et al., 2015*) with UCP1-Cre (*Kong et al., 2014*) transgenic mice to generate *Mpc1*$^{F/F}$::*Ucp1*$^{Cre}$ mice. The conditional deletion of *Mpc1* in brown adipose tissue was confirmed by gene expression analysis (*Figure 3D*). To test whether loss of MPC1 resulted in destabilization of MPC2, we completed western blot analysis and found that MPC2 was also depleted in BAT of *Mpc1*$^{F/F}$::*Ucp1*$^{Cre}$ mice (*Figure 3E* and *Figure 3—figure supplement 1B*). To address whether loss of MPC1 and MPC2 was specific to brown adipose tissue, we also completed western blot analysis on iWAT, and found similar levels of both MPC1 and MPC2 (*Figure 3F* and *Figure 3—figure supplement 1C*). To test whether MPC1 is required for thermogenesis, we completed an acute cold tolerance test at 4°C and measured core body temperature. Upon 5 hours of cold exposure, *Mpc1*$^{F/F}$::*Ucp1*$^{Cre}$ mice had significantly lower core body temperatures when compared to their *Mpc1*$^{F/F}$ littermate controls, suggesting that mitochondrial pyruvate transport is essential for optimal thermogenesis (*Figure 3G*). The cold sensitivity was not due to depletion of glucose, as blood glucose levels were similar between *Mpc1*$^{F/F}$ and *Mpc1*$^{F/F}$::*Ucp1*$^{Cre}$ mice (*Figure 3—figure supplement 1D*).

To determine whether loss of MPC1 led to changes in systemic glucose metabolism, we completed a glucose tolerance test at room temperature (23°C) or with cold (4°C), and found that *Mpc1*$^{F/F}$::*Ucp1*$^{Cre}$ mice had glucose excursion curves that were impaired when compared to their *Mpc1*$^{F/F}$ littermate controls (*Figure 4A*). The loss of MPC1 in BAT did not change body composition of chow-fed mice (*Figure 4—figure supplement 1A*). We also found that CL-316,243 administration resulted in a greater decrease in blood glucose levels in *Mpc1*$^{F/F}$ controls when compared to *Mpc1*$^{F/F}$::*Ucp1*$^{Cre}$ mice (*Figure 4—figure supplement 1B*). In contrast, insulin sensitivity was similar between the two groups as demonstrated by % change in glucose over time (*Figure 4B*). Histological analysis by H and E staining of BAT, iWAT, eWAT, and liver showed little to no differences in tissue morphology between the control and MPC1 null mice (*Figure 4C*). Given that *Mpc1*$^{F/F}$::*Ucp1*$^{Cre}$ mice had a cold sensitive phenotype, we measured gene expression of thermogenic-associated transcripts in BAT, and found that *Mpc1*$^{F/F}$::*Ucp1*$^{Cre}$ mice had reduced expression of *Ucp1*, *Dio2*, *Elovl3*, and *Pparg2* relative to *Mpc1*$^{F/F}$ control mice (*Figure 4D*). No changes were observed in expression of genes involved in de novo lipogenesis and ketolysis (*Figure 4—figure supplement 1C*). To test whether there is compensation for loss of mitochondrial pyruvate uptake, we measured expression of genes that encode for transporters and enzymes involved in fatty acid oxidation. While we observed increased levels of the fatty acid transporter CD36 in *Mpc1*$^{F/F}$::*Ucp1*$^{Cre}$ mice, we saw no differences in *Pnpla2 (Atgl)*, *Cpt1b*, *Cpt2*, or *Agpat2* expression (*Figure 4D*). This suggested that by gene expression, we do not see a compensatory upregulation of fatty acid oxidation in brown adipose tissue of mice lacking MPC1. We also did not find compensatory changes in thermogenic gene expression in iWAT (*Figure 4—figure supplement 1D*). In order to assess whether there is a difference in energy expenditure, food intake, or activity, we placed mice in Columbus Instruments Animal

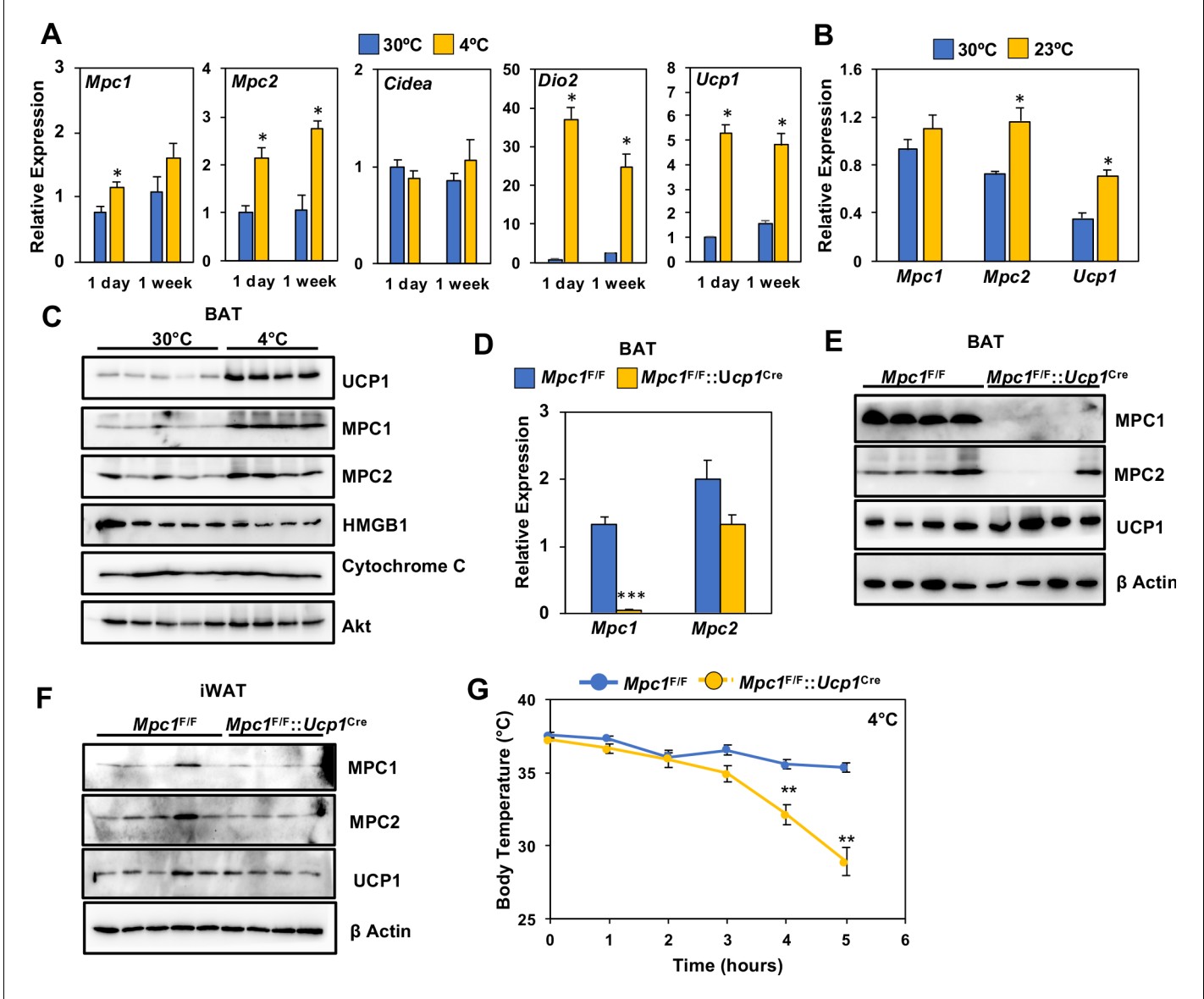

**Figure 3.** Loss of MPC1 in BAT impairs thermogenesis and leads to cold sensitivity. (a) Relative gene expression in brown adipose tissue from mice adapted to 30°C or 4°C for 1 day or 1 week. N = 4–5. (b) Relative gene expression in brown adipose tissue from mice adapted to 30°C or 23°C for 1 week. N = 4–5 (c) Western blot analysis of brown adipose tissue of mice adapted to 30°C or 4°C for 1 week. N = 4–5. (d) Gene expression of MPC1 and MPC2 in brown adipose tissue after 6 hr of cold exposure. N = 7. (e) Western blot analysis of brown adipose tissue and white adipose tissue (f) at 4°C. N = 4. (g) Core body temperature during cold challenge at 4°C. N = 7.

The online version of this article includes the following figure supplement(s) for figure 3:

**Figure supplement 1.** Regulation of *Mpc1* in response to the cold and the conditional deletion of *Mpc1* in brown fat.

Monitoring System (CLAMS), and through continuous monitoring measured energy balance in mice challenged with 6°C. Although we did not find a significant reduction in energy expenditure or change in activity with the loss of MPC1, RER was significantly elevated in *Mpc1*^F/F^::*Ucp1*^Cre^ mice when compared to controls (*Figure 4E, F and G*). Notably, both *Mpc1*^F/F^ and *Mpc1*^F/F^::*Ucp1*^Cre^ mice had reduction in RER, suggesting a metabolic switch toward fat utilization.

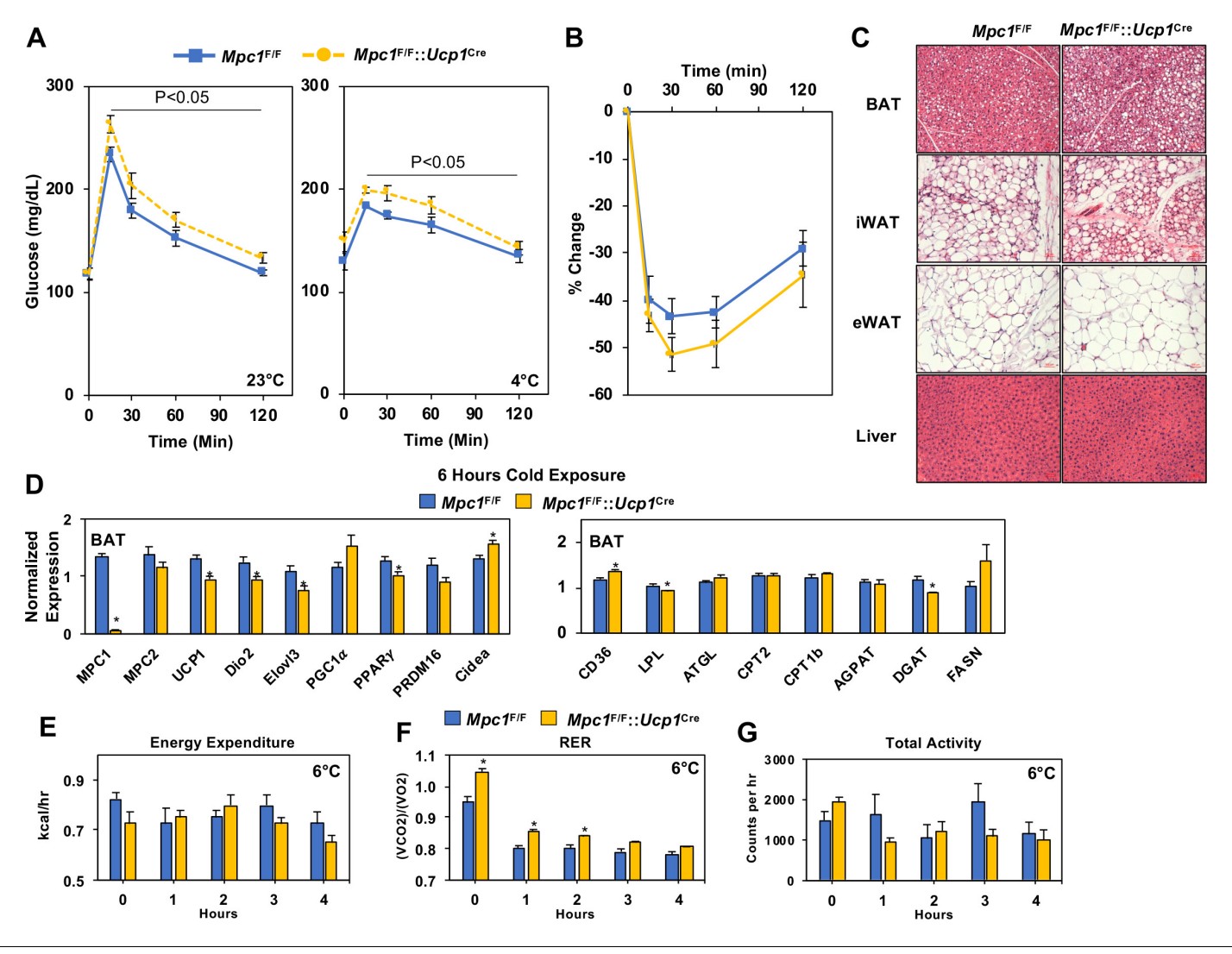

**Figure 4.** Conditional deletion of *Mpc1* in BAT impairs systemic glucose metabolism. (a) Glucose tolerance test at room temperature (23°C) and cold (4°C) in *Mpc1*^F/F and *Mpc1*^F/F::*Ucp1*^Cre 3–4 months old, N = 5. (b) Insulin tolerance test at room temperature (23°C) in *Mpc1*^F/F and *Mpc1*^F/F::*Ucp1*^Cre, 3–4 months old, N = 6. (c). Representative H and E images of BAT, iWAT, eWAT, and liver from *Mpc1*^F/F and *Mpc1*^F/F::*Ucp1*^Cre mice exposed to 4°C for 6 hr. (d) Gene expression in BAT from *Mpc1*^F/F and *Mpc1*^F/F::*Ucp1*^Cre mice exposed to 4°C for 6 hr. N = 6. (e–g) Energy expenditure, RER, and locomotor activity of *Mpc1*^F/F and *Mpc1*^F/F::*Ucp1*^Cre mice at 6°C. N = 4.

The online version of this article includes the following figure supplement(s) for figure 4:

**Figure supplement 1.** Conditional deletion of *Mpc1* in BAT and the impact on body composition and blood glucose.

## Mitochondrial pyruvate transport is required to generate ¹³C-glucose-derived TCA cycle intermediates

While it is well established that cold exposure or CL-316,243 driven stimulation of β3-adrenergic receptor stimulates glucose uptake in brown adipose tissue, the metabolic fate of carbons from glucose has not been fully characterized in brown adipocytes. In order to assess how glucose is metabolized in control cells and those lacking MPC1, we retrovirally expressed MSCV-CreERT2 or empty MSCV control in *Mpc1*^F/F brown preadipocytes to create a tamoxifen inducible knockout system. This allowed us to generate *Mpc1* null cells on day 1 of differentiation as confirmed by western blot (***Figure 5A***) and gene expression analysis (***Figure 5B***). Although *Mpc2* mRNA was not changed (***Figure 5B***), loss of MPC1 led to destabilization and loss of MPC2 (***Figure 5A***). First, we measured the [U-¹³C]-Glucose-derived incorporation into the glycolytic intermediates (***Figure 5C*** and

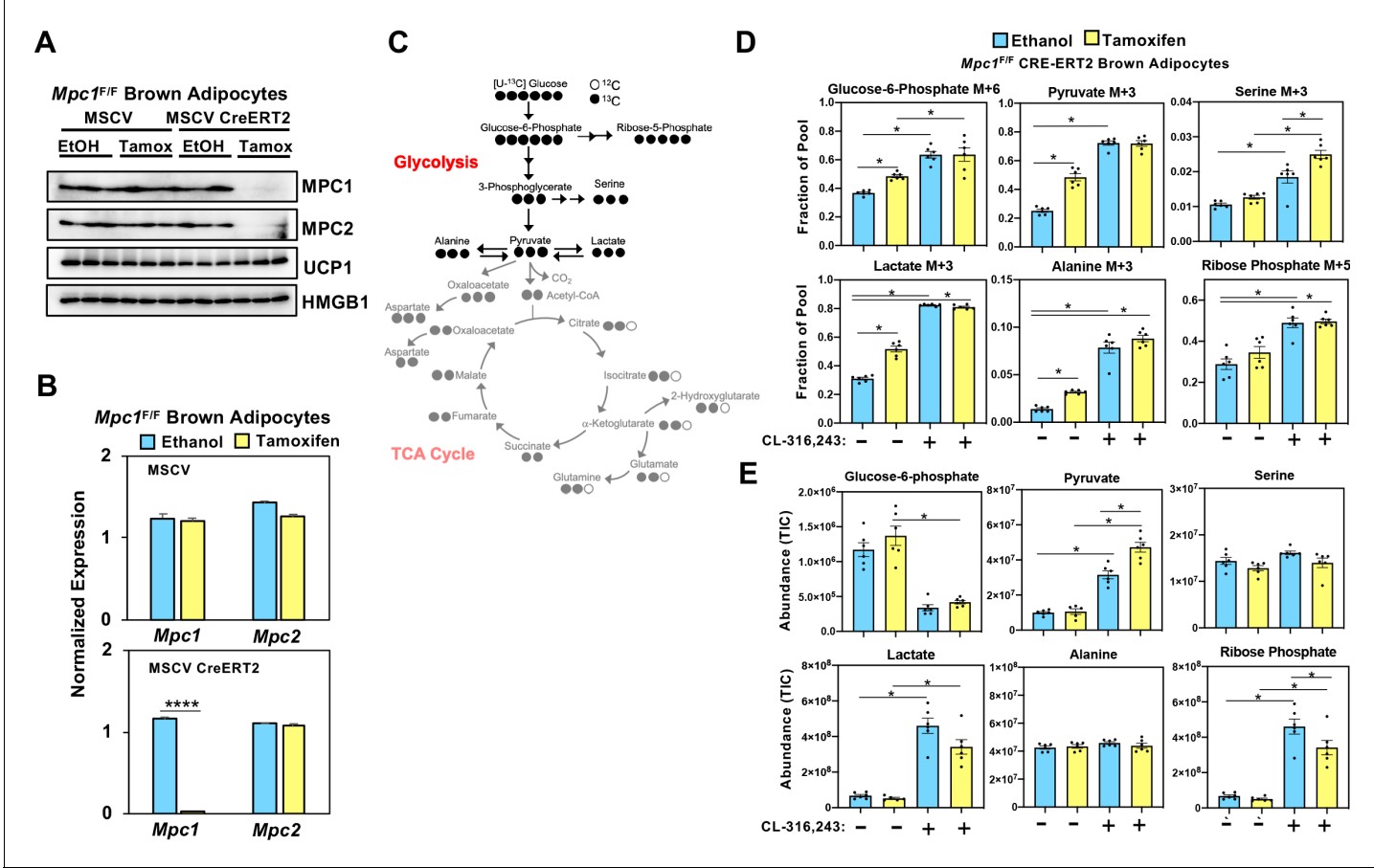

**Figure 5.** Loss of mitochondrial pyruvate carrier does not affect CL-316,243-stimulated increase in $^{13}$C-glycolytic flux. (a) Western blot analysis of differentiated brown *Mpc1*$^{F/F}$ adipocytes expressing pMSCV2 or CreERT2 treated with ethanol or 4-hydroxy tamoxifen. N = 3. (b) Gene expression analysis in differentiated brown *Mpc1*$^{F/F}$ adipocytes expressing pMSCV2 or CreERT2 treated with ethanol or 4-hydroxy tamoxifen N = 3. (c) Atom mapping for [U-$^{13}$C]-glucose tracing incorporation into the glycolytic intermediates. White circles are $^{12}$C atoms. Black circles are $^{13}$C atoms. (d) [U-$^{13}$C]-glucose labeling in *Mpc1*$^{F/F}$ adipocytes expressing CreERT2 treated with ethanol or 4-hydroxy tamoxifen, with/without 100 nM CL-316,243 for 5 hours (N = 6). (e) Steady state levels of glycolytic intermediates in *Mpc1*$^{F/F}$ adipocytes expressing CreERT2 treated with ethanol or 4-hydroxy tamoxifen, with/without 100 nM CL-316,243 for 5 hours (N = 6).

The online version of this article includes the following figure supplement(s) for figure 5:

**Figure supplement 1.** Isotopic labeling of glycolitic intermediates in brown adiopcytes labeled with [U-$^{13}$C]-glucose.

*Figure 5—figure supplement 1A*). After 5 hours of CL-316,243 stimulation, we found extensive M+3 labeling of pyruvate, lactate, serine, alanine, and M+6 labeling of glucose-6-phosphate and M+5 labeling of ribose-5-phosphate in both control and *Mpc1* null cells (*Figure 5D*). Notably, we found greater incorporation of glucose-derived carbons into serine in *Mpc1* null cells treated with CL-316,243. To address whether the pool size changed with CL-326,243 treatment, we measured total abundance of glucose-6-phosphate, ribose-5-phosphate, pyruvate, lactate, alanine, and serine (*Figure 5E*). CL-316,243 treatment led to a dramatic increase in lactate, pyruvate, and ribose-5-phosphate in both control and *Mpc1* null cells. In contrast, CL-316,243 treated MPC1 null brown adipocytes had a greater increase in steady state pyruvate levels (*Figure 5E*). We measured M+3 lactate and M+3 pyruvate in the media to test whether loss of *Mpc1* led to increased [U-$^{13}$C]-Glucose-derived lactate and pyruvate. Upon stimulation with CL-316,243, we found greater levels of M+3 pyruvate and M+3 lactate in the media, with no distinguishable differences between control and knockout cells (*Figure 5—figure supplement 1B*). However, basal levels of M+3 pyruvate and M+3 lactate were elevated in MPC1 null cells. CL-316,243 treatment increased flux through pyruvate dehydrogenase (PDH), which was illustrated by increased M+2 isotopologues of citrate/isocitrate, α-

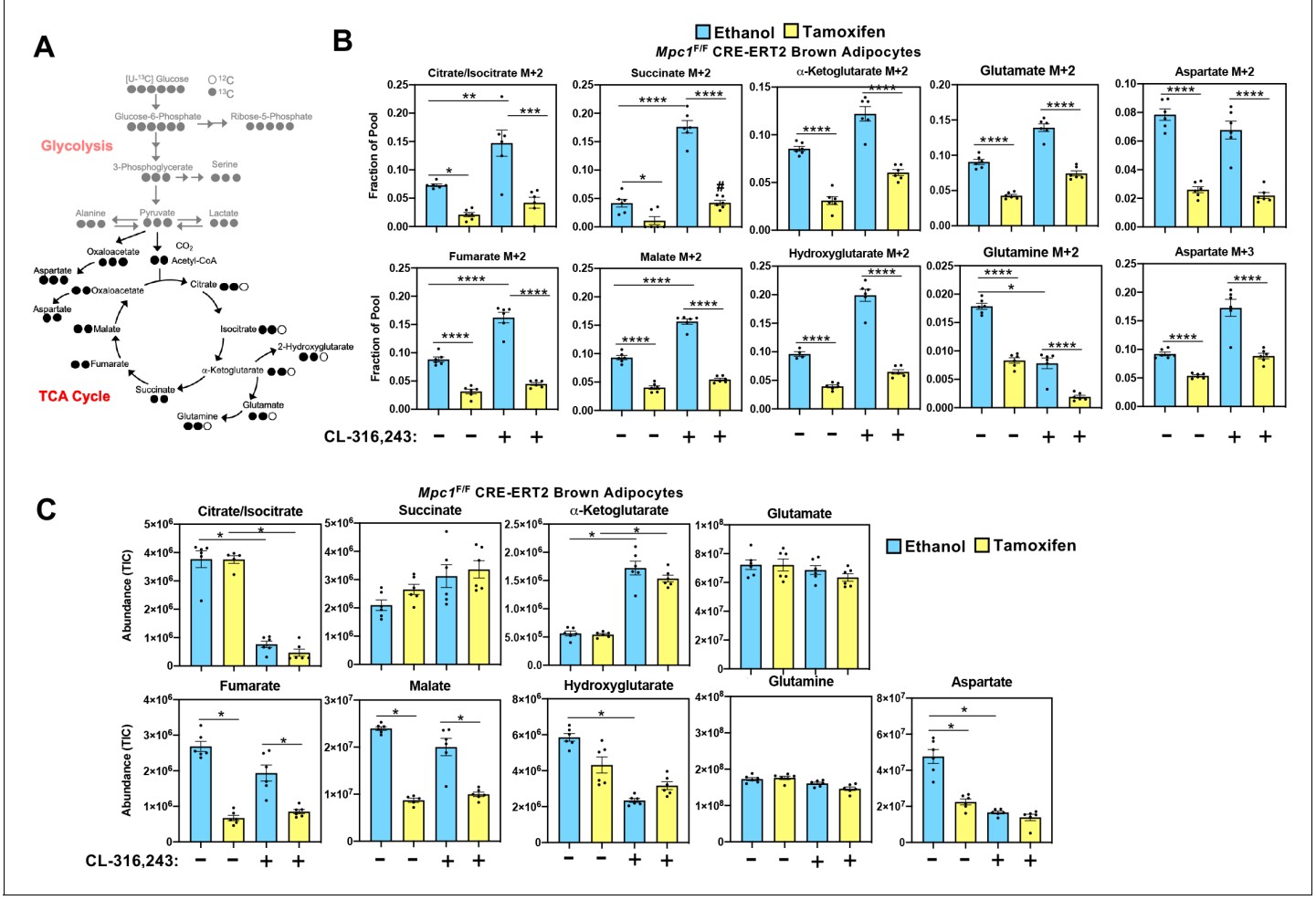

**Figure 6.** Mitochondrial pyruvate transport is required for $^{13}$C-glucose-derived TCA cycle intermediates. (**a**) Atom mapping for [U-$^{13}$C]-glucose tracing incorporation into the TCA cycle intermediates. White circles are $^{12}$C atoms. Black circles are $^{13}$C atoms. (**b**) [U-$^{13}$C]-glucose labeling in *Mpc1*$^{F/F}$ adipocytes expressing CreERT2 treated with ethanol or 4-hydroxy tamoxifen, with/without 100 nM CL-316,243 for 5 hours (N = 6). (**c**) Steady state levels of TCA-cycle intermediates in *Mpc1*$^{F/F}$ adipocytes expressing CreERT2 treated with ethanol or 4-hydroxy tamoxifen, with/without 100 nM CL-316,243 for 5 hours (N = 6).

The online version of this article includes the following figure supplement(s) for figure 6:

**Figure supplement 1.** Loss of Mpc1 in brown adipocytes impairs incorporation of glucose-derived TCA-cycle intermediates.

ketoglutarate, succinate, fumarate, and malate (*Figure 6A and B* and *Figure 6—figure supplement 1A*). To address whether labeling through pyruvate carboxylase was altered with CL-316,243 treatment, we measured M+3 aspartate, a product of M+3 oxaloacetate (*Figure 6B*). We found that M +3 aspartate, increased with CL-316,243 treatment in control cells, while MPC1 null cells had reduced M+3 aspartate. Together we found that loss of MPC1 severely attenuates incorporation of glucose-derived carbons into the TCA cycle, leading to reduced steady state levels of both fumarate and malate (*Figure 6C*).

## MPC1 null brown adipocytes compensate by increasing mitochondrial fatty acid oxidation

Although we found reduced levels of fumarate and malate in MPC1 null brown adipocytes, other TCA cycle intermediates were similar between control and MPC1 null cells (*Figure 6C*). To address whether there was compensation by other fuel sources, we measured fatty acid oxidation using $^{13}$C-U-palmitate complexed with albumin (*Figure 7A*). We found that CL-316,243 treatment increased M

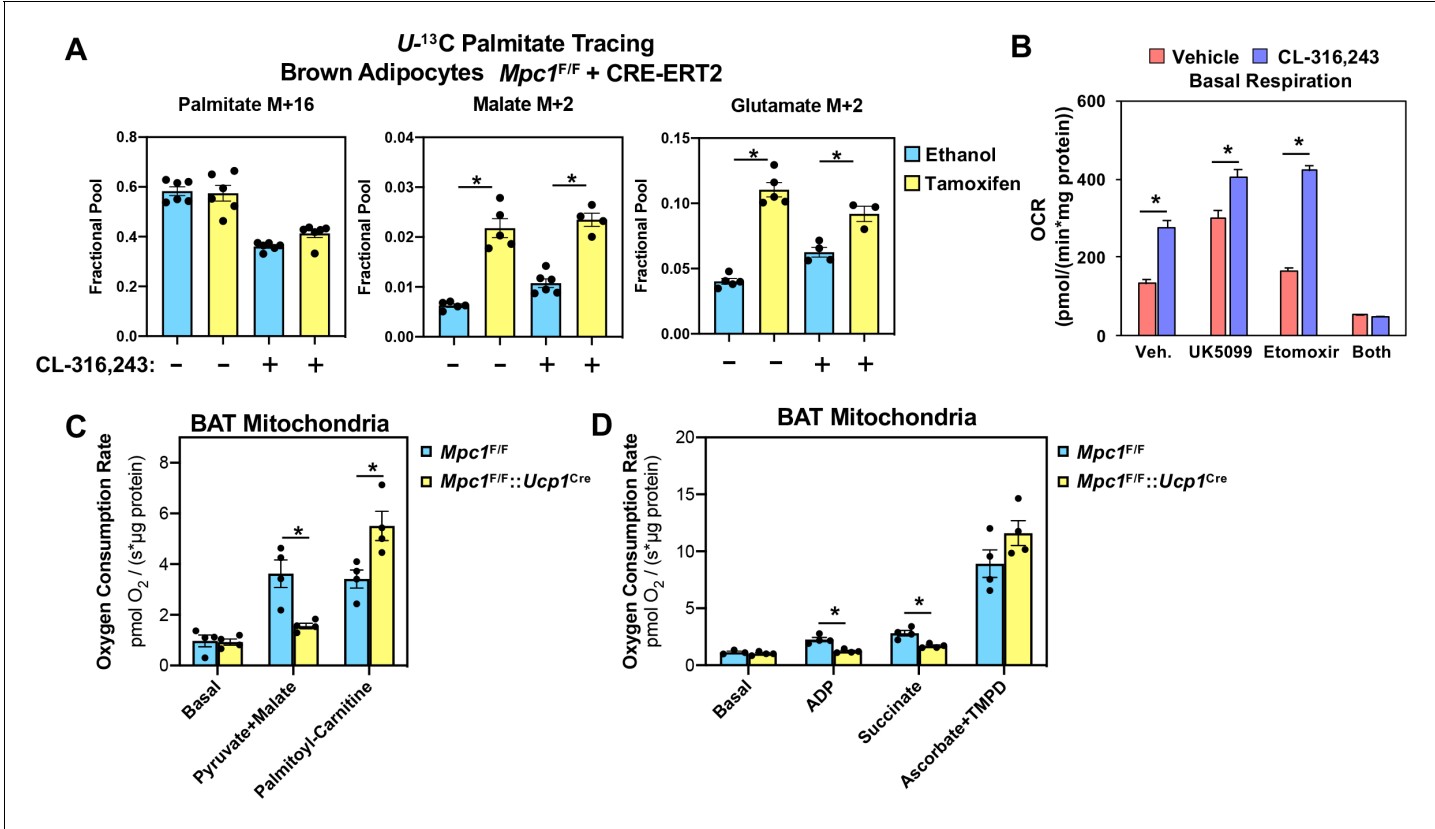

**Figure 7.** Conditional deletion of MPC1 in brown fat leads to compensatory increase in fatty acid oxidation. (a) U-[13]C palmitate-tracing experiments in *Mpc1[F/F]* cells expressing CRE-ERT2. Cells were treated with ethanol or 4-hydroxy tamoxifen, with/without 100 nM CL-316,243 for 5 hours (N = 6). (b) Oxygen consumption rate in differentiated brown adipocytes treated with/without 100 nM CL-316,243 ± vehicle, 10 µM UK5099, 10 µM Etomoxir, or both (N = 10–12). (c) Oxygen consumption in mitochondria isolated from BAT of *Mpc1[F/F]* and *Mpc1[F/F]::Ucp1[Cre]* mice with 5 mM pyruvate and 0.5 mM malate or 5 mM L-Carnitine (N = 4). (d) Oxygen consumption in mitochondria isolated from BAT of Mpc1[F/F] and *Mpc1[F/F]::Ucp1[Cre]* mice with 1 mM ADP, 2 mM ascorbate and 0.5 mM TMPD (N = 4).

The online version of this article includes the following figure supplement(s) for figure 7:

**Figure supplement 1.** Proton leak in response to UK5099, etomoxir, or both.

+2 malate and M+2 glutamate in *Mpc1[F/F]* control cells. However, the loss of MPC1 resulted in greater levels of both M+2 malate and M+2 glutamate in the basal and CL-316,243 treated group. These studies would suggest that loss of MPC1 promotes a compensatory increase in mitochondrial fatty acid oxidation. To test whether acute inhibition of MPC alters respiratory capacity in brown adipocytes, we treated cells with MPC inhibitor UK-5099, and surprisingly found increased oxygen consumption in the basal state and in response to CL-316,243 treatment (*Figure 7B*). To test whether fatty acid oxidation was required for the rise in basal oxygen consumption, we included etomoxir with UK-5099 treatment, which led to a dramatic drop in oxygen consumption. To address whether mitochondrial oxidative capacity was altered in brown adipose tissue, we isolated mitochondria from brown fat of *Mpc1[F/F]* or *Mpc1[F/F]::Ucp1[Cre]* mice. Mitochondria were incubated with defined respiratory substrates, including pyruvate/malate or palmitoyl-carnitine, and found that oxygen consumption increased in *Mpc1[F/F]* brown adipocytes. In contrast, brown fat mitochondria from *Mpc1[F/F]::UCP1[Cre]* mice had reduced oxygen consumption when challenged with pyruvate/malate (*Figure 7C*). However, upon incubation with palmitoyl-carnitine, MPC1 null cells showed a compensatory increase in oxygen consumption rate when provided palmitoyl-carnitine. When ADP or Succinate was added, MPC1 null mitochondria had reduced respiratory capacity relative to *Mpc1[F/F]* controls (*Figure 7D*). To test whether complex IV-dependent respiration was altered, we incubated mitochondria with ascorbate and TMPD, and found similar increase in respiration in *Mpc1[F/F]* and *Mpc1[F/F]::Ucp1[Cre]* brown adipocytes.

## Metabolic profiling in vivo shows increase in Ketogenesis with loss of MPC1 in brown adipose tissue

To understand the systemic metabolic adaptations that occur with the loss of MPC in BAT, we completed metabolomics analysis of serum and BAT in cold challenged $Mpc1^{F/F}$ or $Mpc1^{F/F}$::$Ucp1^{Cre}$ mice. We hypothesized that there may be systemic mechanisms that allow $Mpc1^{F/F}$::$Ucp1^{Cre}$ mice to cope with the loss of MPC during cold stress. Metabolite analysis of serum showed an increase in 3-hydroxybutyrate and adenosine (*Figure 8A*), while metabolite analysis of BAT showed that cold exposed $Mpc1^{F/F}$::$Ucp1^{Cre}$ mice had elevated 3-hydroxybutyrate, 2-hydroxybutyrate, adenosine 5'-monophosphate (AMP), 2-monopalmitoylglycerol, malonic acid, and cis-acotinic acid relative to $Mpc1^{F/F}$ mice (*Figure 8B*). Analysis of the top 25 BAT metabolites showed a significant increase in 3-hydroxybutyrate, while TCA cycle intermediates such as succinic, citric, and isocitric acid were decreased (*Figure 8B*). A list of measured metabolites in BAT and serum of $Mpc1^{F/F}$ or $Mpc1^{F/F}$::$Ucp1^{Cre}$ mice are included in *Supplementary file 5–6*. To test whether ketones were induced with

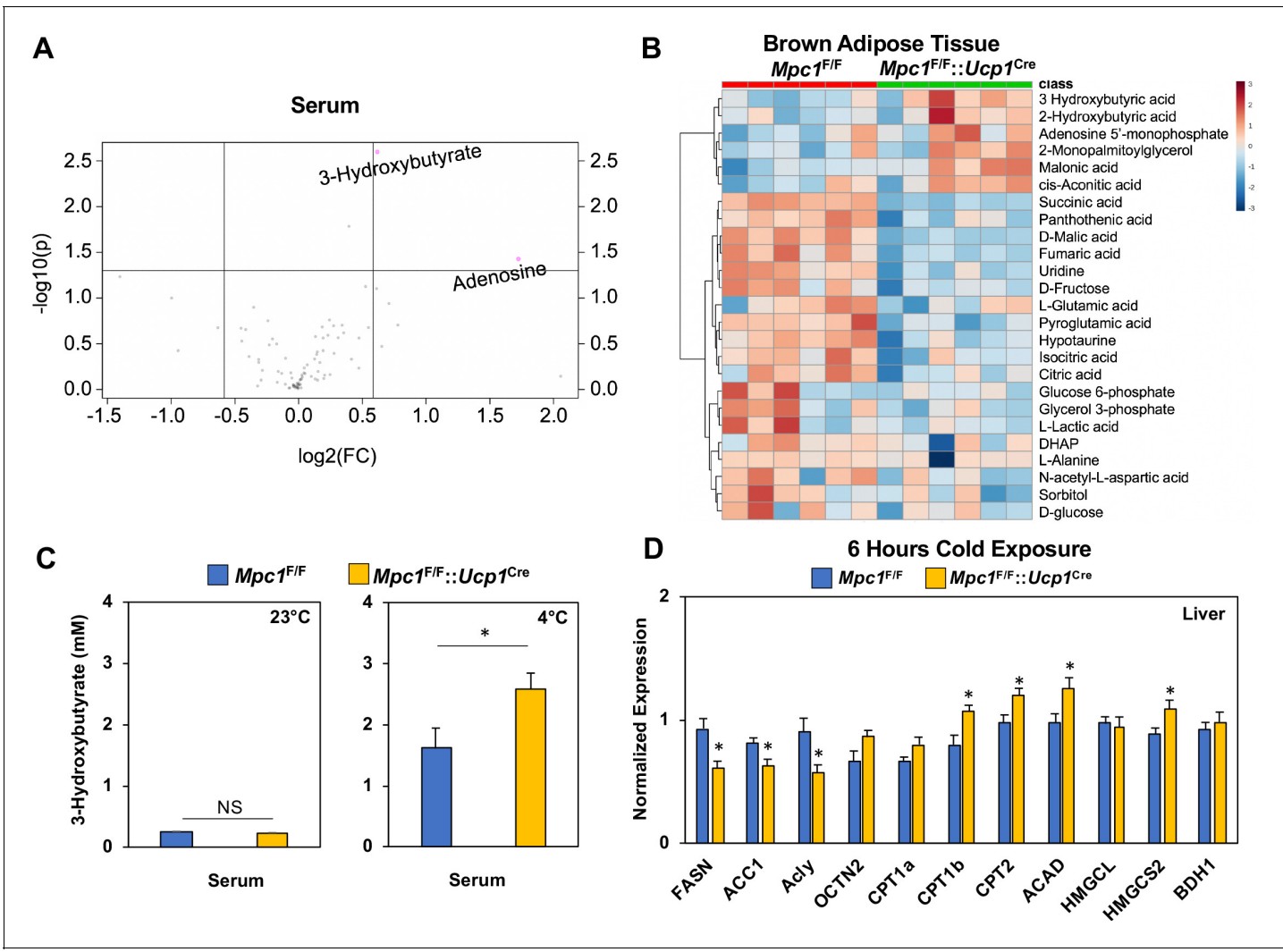

**Figure 8.** Conditional deletion of *Mpc1* in brown fat leads to increased ketogenesis. (a) Volcano plot showing changes in serum metabolites between $Mpc1^{F/F}$ and $Mpc1^{F/F}$::$Ucp1^{Cre}$ mice housed at 4°C for 6 hr. N = 6. (b) Heat map of top 25 metabolites in BAT from $Mpc1^{F/F}$ and $Mpc1^{F/F}$::$Ucp1^{Cre}$ mice housed at 4°C for 6 hr. Dendograms illustrate hierarchical clustering across metabolites (left) and genotypes (top). N = 6. Data was sum normalized, log transformed, and autoscaled. (c) Serum 3-hydroxybutyrate levels from $Mpc1^{F/F}$ and $Mpc1^{F/F}$::$Ucp1^{Cre}$ mice housed at 23°C or 4°C for 6 hr. N = 6. (d) Gene expression analysis of livers from $Mpc1^{F/F}$ and $Mpc1^{F/F}$::$Ucp1^{Cre}$ mice housed at 4°C for 6 hr. N = 6.
The online version of this article includes the following figure supplement(s) for figure 8:

**Figure supplement 1.** Gene expression analysis and serum FFA levels.

cold exposure, we measured serum 3-hydroxybutyrate in *Mpc1*<sup>F/F</sup> and *Mpc1*<sup>F/F</sup>::*Ucp1*<sup>Cre</sup> mice, and found that cold exposure elevated serum 3-hydroxybutyrate levels in *Mpc1*<sup>F/F</sup> control mice (*Figure 8C*). Notably, loss of MPC1 in brown adipose tissue led to blood 3-hydroxybutyrate levels that surpassed those of control mice in the cold (*Figure 8C*). This prompted us to think that liver, being the major ketogenic organ, may be oxidizing more free fatty acids to produce 3-hydroxybutyrate as an alternative fuel for the extrahepatic organs during cold. Therefore, we examined the expression of genes implicated in fatty acid synthesis, oxidation, and ketone body production. While *Fasn*, *Acaca (ACC1)*, and *Acly*, genes involved in fatty acid synthesis, were downregulated in *Mpc1*<sup>F/F</sup>::*Ucp1*<sup>Cre</sup> mice, *Cpt1b*, *Cpt2*, and *Acadm* (fatty acid oxidation genes) were increased, followed by increased levels of *Hmgcs2* which is directly involved in ketogenesis (*Figure 8D*). In contrast, upstream regulators of ketogenesis, including *Ppar* and *Pppargc1a*, were not changed in the livers (*Figure 8—figure supplement 1A*). No notable differences were seen in serum free fatty acids (*Figure 8—figure supplement 1B*), suggesting that activation of ketogenesis is likely contributing to rise in 3-hydroxybutyrate levels. In addition, we did not find changes in expression of ketogenic pathway in iWAT (*Figure 8—figure supplement 1C*). Together, these findings suggest that activation of hepatic ketone production provides an additional compensatory mechanism to counteract the inability to directly oxidize pyruvate in the BAT mitochondria.

## Discussion

There is a prevailing view that BAT relies primarily on free fatty acids as the primary source of energy for brown fat thermogenesis. However, it has been demonstrated in the past that cold activation of BAT leads to utilization of other substrates besides fatty acids, such as glucose, amino acids (*López-Soriano et al., 1988*; *Yoneshiro et al., 2019*) and acylcarnitines (*Simcox et al., 2017*). In this study, we address a fundamental question in BAT thermogenesis: What is the role of glucose oxidation in short-term non-shivering thermogenesis? Is glycolysis or glucose-derived TCA cycle intermediates needed for efficient thermogenesis in BAT? Thus far, there have not been adequate in vivo models to address these questions. In this study, we combined in vitro U-$^{13}$C-glucose tracing experiments with comprehensive in vivo transcriptome and metabolome analysis of activated brown fat to address these questions.

Gene expression profiling of brown adipose tissue showed that mice exposed to short-term cold exposure exhibit evidence of activated cellular respiration, amino acid metabolism, and glucose metabolism. Similar, but distinct findings have been reported with prolonged cold exposure (2–4 days and 10 days respectively)(*Hao et al., 2015*; *Rosell et al., 2014*). In order to see how acute cold exposure affected the metabolome in mice, we followed up these studies by performing GC-MS metabolomics analysis on serum and BAT of mice housed at 30°C, 23°C, and 4°C for 5 hours. This analyses revealed increased branched chain amino acids, ketones, glucose, and TCA cycle metabolites in BAT with decreased temperatures. These results confirmed the previously proposed idea that BAT is a highly metabolically active tissue that upregulates uptake of various fuels to support the energy demand needed to adapt during cold stress. When stimulated with CL-316,243 for 5 hours and given [U-$^{13}$C]-labeled glucose, brown adipocytes significantly upregulated $^{13}$C incorporation into pyruvate, lactate, and TCA cycle intermediates, suggesting that glucose catabolism occurs early in BAT activation. These results are an important complement to recent studies that have described the metabolic response to chronic cold exposure (*Hao et al., 2015*; *Marcher et al., 2015*; *Rosell et al., 2014*). It is not surprising that acute activation of BAT leads to uptake of most substrates available to fuel the heat production process as an initial response to the cold shock. In contrast, cold acclimation or chronic cold exposure, leads to BAT remodeling and adaptive changes such as increased BAT mass, blood flow, and increased mitochondrial number (*López-Soriano et al., 1988*; *Rafael et al., 1985*). Our observation that branched chain amino acids (BCAA) are elevated in BAT with cold exposure is consistent with recent findings highlighting their requirement for optimal thermogenesis (*Yoneshiro et al., 2019*).

Here, we show that mitochondrial pyruvate transport, presumably by its utilization in the TCA cycle, is essential for efficient thermogenesis. In wild type mice challenged with short-term cold exposure, we observe higher levels of MPC1 and MPC2 in BAT compared to that of mice housed at thermoneutrality. We propose that the induction of MPC1 and MPC2 is an adaptive mechanism to increase oxidative capacity during prolonged cold exposure. The inability to directly import pyruvate

into the mitochondria for further oxidation leads to hypothermia, an indication of impaired thermogenesis. This was observed in our $Mpc1^{F/F}::Ucp1^{Cre}$ mouse model where animals had lower core body temperatures during the cold challenge. We also noted small but significant reductions in thermogenic gene expression ($Ucp1$, $Dio2$, $Elovl3$, $Pparg$), but there were no compensatory changes in BAT expression of genes required for fatty acid oxidation. However, we did note that loss of MPC leads to upregulation of $Cd36$, which may drive increased fatty acid uptake during the cold (*Bartelt et al., 2011*). An obvious alternative source of energy is fatty acids, therefore we measured fatty acid oxidation in brown adipocytes and isolated mitochondria. As noted in the intestine lacking MPC1, brown adipocytes have increased fatty acid oxidation (*Bensard et al., 2020*). These effects highlight that the inability to transport pyruvate, leads to compensatory metabolic programming towards fatty acid oxidation. The shift towardfatty acid oxidation was supported both by our tracer studies and measurement of mitochondrial oxygen consumption when palmitoyl-carnitine is provided as a substrate. In addition, there may be compensation through glutamine oxidation, transamination of alanine to pyruvate in the mitochondria, glutamine anaplerosis via glutamate to α-ketoglutarate by glutaminase and glutamate dehydrogenase enzymes, or conversion of glutamine-derived malate to pyruvate by mitochondrial malic enzyme (*Bender and Martinou, 2016*; *Gray et al., 2015*; *McCommis et al., 2015*; *Schell et al., 2014*; *Vacanti et al., 2014*; *Yang et al., 2014*).

One striking feature observed with cold adaptation in $Mpc1^{F/F}::Ucp1^{Cre}$ mice and their littermate controls was elevated ketone levels in the blood. $Mpc1^{F/F}::Ucp1^{Cre}$ mice had significantly elevated serum 3-hydroxybutyrate levels after 6 hr of cold challenge, but there were no measurable differences between the two groups after 6 hr at room temperature. These changes were accompanied by elevated 3-hydroxybutyrate levels in the BAT. Ketogenesis occurs primarily in the liver during exercise or prolonged fasting, and more recently was found to be a cold-induced metabolite (*Newman and Verdin, 2014b*; *Newman and Verdin, 2014a*; *Wang et al., 2019*). Ketones can be exported to extrahepatic tissues for further oxidation as they are rich energy sources. When we measured ketogenic gene expression in the liver, we found that $Mpc1^{F/F}::Ucp1^{Cre}$ mice had significantly increased $Hmgcs2$ levels compared to control mice. Together with serum and BAT metabolomics data this suggest that $Mpc1^{F/F}::Ucp1^{Cre}$ mice compensate by activating ketone production. At first, we speculated that 3-hydroxybutyrate is utilized by BAT of $Mpc1$-deficient mice to compensate for the inability to oxidize pyruvate. However, in order for ketones to be catabolized in peripheral tissues they have to utilize OXCT1 for import and succinyl-CoA to donate coenzyme-A. In the BAT metabolomics analysis, $Mpc1^{F/F}::Ucp1^{Cre}$ mice had lower levels of TCA cycle intermediates compared to their littermate controls, including succinic acid, citric acid, and malic acid. This would suggest that oxidative metabolism is limited in the absence of MPC1. Further, this poses a question of why would $Mpc1^{F/F}::Ucp1^{Cre}$ mice make more 3-hydroxybutyrate and what role it might have in these mice? One likely explanation is that BAT utilizes ketones for thermogenesis. Alternatively, ketones can promote energy expenditure, mitochondrial biogenesis, and stimulate the expression of $Ucp1$ in WAT (*Srivastava et al., 2012*).

Taken together, our studies aimed to gain a better understanding of the metabolic fate of glucose in BAT during short-term cold exposure. Here, we report a novel mouse model of $Mpc1$ loss in brown adipocytes that allowed us to assess the importance of efficient pyruvate import and oxidation for thermogenesis. Understanding the metabolic pathways and key metabolites that are upregulated in brown fat during cold exposure could provide new therapeutic targets to treat metabolic disorders such as obesity and diabetes.

## Materials and methods

### Animals

All procedures were approved by the Institutional Animal Care and Use Committee (IACUC) of University of Utah. Mice were housed at 22–23˚C using a 12 hr light/12 hr dark cycle. Animals were maintained on a regular Chow diet (2920x-030917M). Mice had ad libitum access to water at all times. Food was only withdrawn during experiments. C57BL/6J male mice at 3 months of age were purchased from Jackson Laboratories. C57BL/6J $Mpc1^{F/F}$ mice were generated as previously described (*Birsoy et al., 2015*). Floxed mice were crossed with C57BL/6J $Ucp1^{Cre}$ (Jax #024670)

mice to generate conditional deletion of *Mpc1* in brown adipocytes. Floxed Cre-negative littermates were used as controls. The age of mice used for all the studies were 12–20 weeks old. No animals were excluded from any experiments.

## Cold exposure

For short-term cold exposure studies (5–6 hr) mice were singly housed without food, nor bedding, with free access to water. Starting at T0 mice were placed at either 30°C (thermoneutrality), 23°C (room temperature), or 4–6°C (cold exposure) for 6 hr. Body temperatures were taken once every hour with a physitemp A590 rectal probe using an Oaklon Thermocouple digital thermometer. For long-term cold exposure studies (1 week) mice were individually housed, with bedding and ad libitum access to food and water.

## Glucose tolerance and insulin tolerance tests

For glucose tolerance test 12 weeks old mice were fasted for 6 hr and then administered 1 g/kg of body weight of glucose by intraperitoneal injection. For insulin tolerance test non-fasted mice were administered 0.75 units/kg of body weight of insulin. Glucose levels were measured by tail vein using Contour next one glucometers at the indicated time points.

## Metabolic cages

Food and water intake, energy expenditure and ambulatory activity were measured by using Comprehensive Lab Animal Monitoring System (CLAMS) (Columbus Instruments) through the University of Utah Metabolic Phenotyping core. Mice were single housed in metabolic cages with ad libitum access to food and water on a 12 hr light/12 hr dark cycle. Mice were single housed in metabolic cages with no food and with free access to water. Temperature was was set at 6°C and measurements were obtained for a period of 4 hours. Energy expenditure was calculated as a function of oxygen consumption and carbon dioxide production in the CLAMS cages.

## CL-316,243 Treatment

CL-316,243 (1 mg/kg body weight; Sigma) or a vehicle control sterile PBS pH 7.4 was injected intraperitoneally. After drug or vehicle were administered, glucose levels were measured once every hour for 6 hr by tail vein using Contour next one glucometer. During this time mice were single housed at 23°C, without food but water was readily available.

## Cell culture

Brown preadipocytes were isolated from 6-week-old MPC1 F/F mice (*Rodriguez-Cuenca et al., 2007*). Intrascapular BAT was removed, minced, and digested in buffer containing 1% collagenase, DMEM (Cat# 11995073, Invitrogen Life) and antibiotics-50 IU Penicillin/mL and 50 μg Streptomycin/mL (Cat# 15140122, Invitrogen Life) plus Primocin 100 μg/mL (Cat# ANT-PM-2, Invivogen). Samples were incubated in the shaking water bath at 37°C for 45 min after which they were allow to cool on ice for 20 min. Infranatant was filtered through a 100 μm filter and centrifuged for 5 min at 500xg. The digestion buffer was removed and pellet was washed twice with DMEM with antibiotics. After the last spin pellet was resuspended in 1 mL of DMEM containing 10% FBS (Cat# FB-01, Omega Scientific, Inc) and antibiotics. Cells were then plated into a six-well plate and the next day they were immortalized by retroviral expression of SV40 Large T-antigen (Cat# 13970, Adgene) using hygromycin for selection. For MPC1 null studies, stable expression of CreERT was generated using pMSCV CreERT2 retroviral vector (Cat# 22776, Adgene) with puromycin selection marker. Cells are routinely tested for mycoplasma prior to experimentation. For gene expression experiments, the cells were plated in 12-well plates (75,000 cells/well) in DMEM containing 10%FBS, 1nM T3 (Cat# T6397, Sigma), and 20 nM insulin (Cat# 91077C, Sigma). Upon confluency cells were given differentiation cocktail containing 10%FBS, 1nM T3, 20 nM insulin, 1 μM rosiglitazone (Cat#71740, Cayman Chemical), 0.5 μM dexamethasone (Cat# D4902, Sigma), 0.5 mM isobutylmethylxanthine (Cat# I5879, Sigma), and 0.125 mM indomethacin (Cat# I7378, Sigma). After 1 day of differentiation 100 nM 4-hydroxy-tamoxifen (Cat# 3412, Tocris) was added to knock out MPC1 gene or DMSO (Cat# D2650, Sigma) was added as a control. After 2 days of differentiation, media was changed to DMEM

containing 10% FBS, 1nM T3, 20 nM insulin, and 1 µM rosiglitazone. Cells were harvested on day 9 of differentiation for different experimental analyses.

## Brown adipocyte U-$^{13}$C glucose and U-$^{13}$C palmitate labeling

Cells were plated in a 6-well plate at a seeding density of 200,000 cells/well. On day 8 of differentiation cells were washed twice with 1XPBS and media was changed to high-glucose DMEM (Cat# 11995073, Thermo Fisher) containing 10% FBS overnight. The next day this media was removed and cells were washed twice with 1X PBS. They were incubated in a glucose/phenol red/glutamine-free DMEM (Cat# A14430-01, Thermo Fisher) with added 5.5 mM glucose (Cat# G8270, Sigma), Gluta-Max(Cat# 35050061, Thermo Fisher), and MEM Non-Essential Amino Acid Solution (Cat# 11140050, Thermo Fisher). Cells were allowed to equilibrate for 4 hr before the media was changed to the same composed DMEM but this time containing 5.5 mM U-$^{13}$C D-Glucose (Cat# CLM-1396–5, Cambridge Isotopes). For U-$^{13}$C palmitate labeling same composed media containing 5.5 mM glucose was used with added 150 µM U-$^{13}$C Sodium palmitate (CLM-6059–1, Cambridge Isotopes) conjugated to fatty-acid-free BSA (Cat# 700–107P, Gemini Bio Products) and 1 mM Carnitine (Cat# C0823, Sigma). In both experiments, cells were stimulated with 100 nM CL-316,243 or vehicle for 5 hours. Before harvesting the cells 1 mL od media was taken and centrifuged at 21,000xg for 10 min at 4°C. 40 µL of supernatant were added to 160 µL of ice-cold 80% methanol for metabolic tracing analysis. The remaining media was removed and cells were harvested by addition of 200 µL of −80° C chilled buffer containing 20% water and 80% methanol (Cat# AA47192M6, Fisher Scientific). Lysed cells were kept on dry ice for 5 min before collection. Samples were spun down as before and 100 µL of supernatant was directly used for metabolic tracer analysis.

### Measure of oxygen consumption

Oxygen consumption rate was measured using a Seahorse XF96e analyzer through the University of Utah Metabolic Phenotyping core. 35,000 differentiated brown adipocytes were plated in each well of a XF 96-well cell culture plate in 100 µL of DMEM culture media and allowed to attach overnight. Cells were pre-treated overnight in vehicle or 10 µM UK5099 and incubated at 37°C in 5% CO$_2$. Next day the culture media was replaced with standard assay media (DMEM, 25 mM glucose, 1 mM pyruvate, 2 mM glutamine, pH 7.4). Cells were pretreated with 10 µM Etomoxir for 15 min and activated with/without 100 nM CL-316,243. Cells were run on a XF96e analyzer for a Mito Stress Test using manufacturers protocol and standard drug concentrations (Oligomycin 2.5 µM, FCCP 2 µM, Rotenone 0.5 µM, and Antimycin A 0.5 µM). Assay protocol was standard (three measurements per phase, acute injection followed by 3 min of mixing, 0 min waiting, and 3 min measuring). Data was normalized to total cellular protein levels per well (ThermoFisher BCA Kit cat #23227).

### Mitochondrial isolation and measure of mitochondrial oxygen consumption

Mitochondria were isolated from MPC1$^{F/F}$ and MPC1$^{F/F\ UCP1Cre}$ mice. BAT was excised and placed in ice-cold mitochondrial isolation media (MIM) consisting of 300 mM sucrose, 10 mM HEPES, 1 mM EGTA, pH 7.2 and minced. The tissue was then gently homogenized and centrifuged for 10 min, 4° C, at 10,000 rcf. The floating lipid layer and supernatant were then aspirated and the pellet was resuspended in MIM + 1 mg / mL BSA. To remove cellular debris, samples were then split into two tubes, centrifuged for 5 min, 4°C at 200 rcf and the supernatant was saved (discarding the pellet) two consecutive times. Samples were then centrifuged for 10 min, 4°C at 10,000 rcf to pellet the mitochondria. Finally, samples were resuspended in MIM and an aliquot was used to determine protein content by BCA Assay.

25 µg of mitochondria were loaded in triplicate into the Oroboros O2K High-Resolution respirometer in 2.1 mL of Buffer Z (105 mM MES Potassium Salt, 30 mM KCl, KH2PO4 10 mM, MgCl2-6H2O 5 mM, Fatty-acid free BSA 0.5 mg/ml). Respiratory oxygen flux was measured in real time and reported as pico moles O2 consumed per second per mg mitochondria. 5 mM pyruvate and 0.5 mM malate were added followed by 5 mM L-Carnitine. In a separate experiment, Complex I activity was measured by the addition of 5 mM pyruvate, 0.5 mM malate, and 1 mM ADP. Complex II respiration was then tested by the addition of 5 mM Succinate. Finally, Complex IV was tested using 2 mM ascorbate and 0.5 mM N,N,N,N-tetramethyl-p-phenylenediamine (TMPD).

## FFA measurement

Free fatty acids were measured from the blood serum of MPC1 null mice and their littermate controls that were housed at room temperature or challenged by cold for 6 hr. 10 μL of the serum was used for analysis using commercial kit (Cat# MAK044-1KT, Sigma) according to the manufacturer instructions.

## Metabolite extraction

In order to extract metabolites from the tissue, each sample was transferred to 2.0 ml ceramic bead mill tubes (bioExpress). Each sample received 450 ul of 90% cold methanol in diH2O for every 25 mg of tissue. The samples were then homogenized in an OMNI Bead Ruptor 24. Homogenized samples were then incubated at −20 ˚C for 1 hr. D4-succinic acid (Sigma 293075) was added to each sample as an internal standard. After incubation, all the samples were centrifuged at 20,000 x g for 10 min at 4 ˚C. 450 ul of supernatant was then transferred from each bead mill tube into a labeled, fresh micro centrifuge tube where another internal standard d27-myristic acid (CDN Isotopes: D-1711). Samples were then dried *en vacuo.* For metabolite extraction from serum, 90% methanol in diH2O containing d4-succinic acid was added to each sample to give a final methanol concentration of 80%. Samples were vortexed and incubated at −20 ˚C for 1 hr. After incubation, all samples were centrifuged at 20,000 x g for 10 min at 4 ˚C. Another internal standard, d27-myristic acid (CDN Isotopes: D-1711), was added to each sample. Process blanks were made using the extraction solvent and went through the same process steps as the real samples. The samples were then dried *en vacuo.*

## GC-MS analysis of metabolites

All GC-MS analysis was performed with an Agilent 7200 GC-QTOF and an Agilent 7693A automatic liquid sampler. Dried samples were suspended in 40 μL of a 40 mg/mL O-methoxylamine hydrochloride (MOX) (MP Bio #155405) in dry pyridine (EMD Millipore #PX2012-7) and incubated for 1 hr at 37˚C in a sand bath. 25 μL of this solution was added to auto sampler vials. 60 μL of N-methyl-N-trimethylsilyltrifluoracetamide (MSTFA with 1%TMCS, Thermo #TS48913) was added automatically via the auto sampler and incubated for 30 min at 37˚C. After incubation, samples were vortexed and 1 μL of the prepared sample was injected into the gas chromatograph inlet in the split mode with the inlet temperature held at 250˚C. A 5:1 split ratio was used for analysis for the majority of metabolites. Any metabolites that saturated the instrument at the 5:1 split were analyzed at a 50:1 split ratio. The gas chromatograph had an initial temperature of 60˚C for one minute followed by a 10 ˚C/min ramp to 325˚C and a hold time of 10 min. A 30-meter Agilent Zorbax DB-5MS with 10 m Duraguard capillary column was employed for chromatographic separation. Helium was used as the carrier gas at a rate of 1 mL/min. Below is a description of the two-step derivatization process used to convert non-volatile metabolites to a volatile form amenable to GC-MS. Pyruvic acid is used here as an example.

## Analysis of GC-MS metabolomics data

Data was collected using MassHunter software (Agilent). Metabolites were identified and their peak area was recorded using MassHunter Quant. This data was transferred to an Excel spread sheet (Microsoft, Redmond, WA). Metabolite identity was established using a combination of an in-house metabolite library developed using pure purchased standards, the NIST library and the Fiehn library. There are a few reasons a specific metabolite may not be observable through GC-MS. The metabolite may not be amenable to GC-MS due to its size, or a quaternary amine such as carnitine, or simply because it does not ionize well. Metabolites that do not ionize well include oxaloacetate, histidine and arginine. Cysteine can be observed depending on cellular conditions. It often forms disulfide bonds with proteins and is generally at a low concentration. Metabolites may not be quantifiable if they are only present in very low concentrations.

## LC-MS analysis of polar metabolites

Extracted polar metabolite samples were analyzed by LC-MC. Separation was achieved by hydrophilic interaction liquid chromotograhpy (HILIC) using a Vanquish HPLC system (ThermoFisher Scientific). The column was an Xbridge BEH amide column (2.1 mm x 150 mm, 2.5 μM particular size, 130

Å pore size, Waters Co.) run with a gradient of solvent A (20 mM ammonium hydroxide, 20 mM ammonium acetate in 95:5 acetonitrile:Water, pH 9.5) and solvent B (100% acetonitrile) at a constant flow rate of 150 uL/min. The gradient function was: 0 min, 90% B; 2 min, 90% B; 3 min, 75% B; 7 min, 75% B; 8 min, 70% B; 9 min, 70% B; 10 min, 50% B; 12 min, 50% B; 13 min, 25% B; 14 min, 25% B; 16 min, 0% B; 20.5 min, 0% B; 21 min; 90% B; 25 min, 90% B. Autosampler temperature was 4°C, column temperature 30°C and injection volume 2 μL. Samples were injected by electrospray ionization into a QExactive HF orbitrap mass spectrometer (ThermoFisher Scientific) operating in negative ion mode with a resolving power of 120,000 at m/z of 200 and a full scan range of 75–1000. Data were analyzed using the MAVEN software package and specific peaks assigned based on exact mass and comparison with known standards (*Melamud et al., 2010*). Extracted peak intensities were corrected for natural isotopic abundance (*Su et al., 2017*).

## Gene expression

RNA was isolated from differentiated brown adipocytes or from brown adipose tissue or white adipose tissue using Trizol reagent (Cat# 15596018, ThermoFisher). Tissue samples were homogenized with a TissueLyzer II (Qiagen). Isolated RNA was reverse transcribed using SuperScript VILO Mastermix (Cat# 11755500, ThermoFisher). Gene expression was quantified using Quant Studio 6 Flex Real-Time PCR instrument, 384-well (Applied Biosystems by Invitrogen) with KAPA SYBR FAST qPCR 2x Master Mix Rox Low (Cat# KK4621, Kapa Biosystems). Relative mRNA expression of indicated transcripts was normalized to expression of the housekeeping gene RPS3. Primers were designed using Universal Probe Library (Roche) or qPrimer Depot. A list of primer sequences can be found in *Supplementary file 7*.

## Western blots

Cells were lysed using Radioimmunoprecipitation assay (RIPA) buffer (Boston Bioproducts, Inc) plus protease inhibitor cocktail (Cat# 04693124001, Sigma Aldrich) and phosphatase inhibitor cocktail (Cat# 78428, ThermoFisher). Lysates were passaged through a 25-gauge needle 10 times. Snap-frozen tissues were homogenized using a TissueLyzer II (Qiagen) in the same lysis buffer. Cell/tissue lysates were centrifuged twice at 13,000 rpm at 4°C for 10 min. Lipid layer was removed after each centrifugation. Protein concentrations were measured using Pierce BCA Protein Assay Kit (Cat# 23225, Thermo Fisher). 20 μg of total protein was denatured using Laemmli buffer and samples were heated at 50°C for 10 min. Protein was loaded onto 10% acrylamide/bisacrylamide gels and transferred to a nitrocellulose membrane (GE Healthcare) for 60 min at 100 V for detection with the indicated antibodies. Briefly, membranes were blocked in 5% milk/PBST for 1 hr and then incubated with primary antibodies (1:1000 dilution) in 5% BSA/PBST overnight at 4°C. Horse radish peroxidase-conjugated secondary antibodies (1:4000 dilution) were given for 1 hr. Western blots were developed using WesternSure Premium Chemiluminescent substrate (Cat# C807723-02, LI-COR Biosciences) and detected by ChemiDoc MP Imaging System (BioRad).

## Antibodies and reagents

MPC1 (14462), MPC2 (46141), β-Actin (4970), Akt (9272) were purchased from Cell Signaling Technologies, UCP1 (AB10983), Cytochrome C [7H8.2C12] (AB13575), HMGB1 (AB18256) were purchased from Abcam. 4-hydroxy-tamoxifen (4-OHT) and UK5099 were purchased from Tocris. CL-316,243 (C5796) was purchased from Sigma. U-$^{13}$C D-Glucose (CLM-1396–5) and U-$^{13}$C Sodium palmitate (CLM-6059–1) were purchased from Cambridge Isotopes. Sodium palmitate (P9767) was purchased from Sigma Aldrich. DL-[1-$^{14}$C] 3-hydroxybutyric acid sodium salt (ARC1455) was purchased from American Radiolabeled Chemicals. DL-β-Hydroxybutyric acid sodium salt (H6501) was purchased from Sigma.

## Quantification and statistical analysis

Assessment of metabolomics using hierarchical clustering was performed using MetaboAnalyst 3.0 (*Xia and Wishart, 2016*). The data was interquartile range filtered, sum normalized, log2 transformed and autoscaled. Comparison of differentially abundant plasma or BAT metabolites from 3-month-old mice in 30°C, 23°C, or 4°C was performed in MetaboAnalyst 3.0 by using 1-way ANOVA

analysis followed by Tukey's HSD post hoc test. All other data are presented as mean ± SEM and Student's t-test was used to determine significance, unless otherwise stated.

## RNA sequencing and data processing

We used the standard procedure of Qiagen RNeasy kit to extract total RNA from BAT of mice. The RNA library for sequencing was prepared using TruSeq Stranded mRNA Library Prep Kit (Illumina, San Diego, CA) and rRNA was removed by Ribo-Zero following the protocol provided by the manufacturer. The final libraries were normalized in preparation pooling by Kapa Library Quantification Kit for Illumina Platforms and the libraries were sequenced with the Illumina HiSeq 2000 sequencing platform within a lane for all six samples. For RNA-seq data process, we used Rsubread (Bioconductor release 3.8) [23558742] to align sequence reads to reference genome and used edgeR [22287627] and Limma [25605792] R packages (Bioconductor release 3.8) to normalize gene expression level to log2 transcripts per million (TPM) [22872506]. We aligned sequence reads to GRCh38 human genome reference sequence and mapped the aligned sequences to Ensembl or Entrez Gene IDs. After normalization for every sample, we used young room temperature (five mice) and cold room exposed (five mice) samples in this study. The raw RNA-seq data files and normalized expression profile data is available through GEO (GSEOOOOOO).

## Clustering analysis and Gene Set Enrichment Analysis (GSEA)

We removed genes of which expression level is zero across all samples and explored the expression clusters between young room temperature and cold room exposed groups. We performed unsupervised hierarchical clustering analysis and Principal Component Analysis (PCA). We used Euclidean distance metric in hierarchical clustering, and the first three components in PCA. Furthermore, we validated this result with the supervised learning method, Random Forest. To identify biological processes whose expression differed between the clusters, we ran GSEA using Gene Ontology biological process (version 4.0) gene signatures [16199517]. In this analysis, we used all genes and calculated p-values by permuting the class labels 1000 times. Gene sets with a false discovery rate (FDR) q-value <0.25 were considered significant. To visualize relationships among the top-performing gene signatures, we used EnrichmentMap [22962466].

## Additional information

### Funding

| Funder | Grant reference number | Author |
|---|---|---|
| National Institutes of Health | 1R01DK103930 | Claudio J Villanueva |

The funders had no role in study design, data collection and interpretation, or the decision to submit the work for publication.

### Author contributions

Vanja Panic, Conceptualization, Data curation, Formal analysis, Supervision, Funding acquisition, Investigation, Visualization, Methodology, Project administration; Stephanie Pearson, Data curation, Formal analysis, Investigation, Visualization, Methodology; James Banks, Sanghoon Lee, James Cox, Data curation, Formal analysis, Investigation; Trevor S Tippetts, Will L Holland, Investigation, Methodology; Jesse N Velasco-Silva, Judith Simcox, Gisela Geoghegan, Tyler van Ry, Jared Rutter, Investigation; Claire L Bensard, Formal analysis, Investigation; Scott A Summers, Methodology, Project administration; Gregory S Ducker, Formal analysis, Investigation, Methodology; Claudio J Villanueva, Conceptualization, Resources, Formal analysis, Supervision, Funding acquisition, Investigation, Project administration

### Author ORCIDs

Trevor S Tippetts (iD) http://orcid.org/0000-0002-1419-7057
Jared Rutter (iD) http://orcid.org/0000-0002-2710-9765
Claudio J Villanueva (iD) https://orcid.org/0000-0002-9731-7463

## Ethics

Animal experimentation: This study was performed in strict accordance with the recommendations in the Guide for the Care and Use of Laboratory Animals of the National Institutes of Health. All of the animals were handled according to approved institutional animal care and use committee (IACUC) protocols (#18-08004) of the University of Utah. The protocol was approved by the Committee on the Ethics of Animal Experiments of the University of Utah.

## Decision letter and Author response

Decision letter https://doi.org/10.7554/eLife.52558.sa1
Author response https://doi.org/10.7554/eLife.52558.sa2

---

# Additional files

## Supplementary files

• Supplementary file 1. List of 1907 genes up-regulated in response to the cold.

• Supplementary file 2. List of 3273 genes down-regulated in response to the cold.

• Supplementary file 3. Metabolite analysis in brown adipose tissue at 30°C, 23°C, and 4°C.

• Supplementary file 4. Metabolite analysis of serum at 30°C, 23°C, and 4°C.

• Supplementary file 5. List of metabolites from serum of MPC1 F/F and MPC1 F/F::UCP1-Cre mice housed at 4°C for 6 hr.

• Supplementary file 6. List of metabolites from BAT of MPC1 F/F and MPC1 F/F UCP1 Cre mice housed at 4°C for 6 hr.

• Supplementary file 7. Real-time PCR primer list.

• Transparent reporting form

## Data availability

RNA sequencing data will be deposited in GEO under accession codes GSE135391.

The following dataset was generated:

| Author(s) | Year | Dataset title | Dataset URL | Database and Identifier |
|---|---|---|---|---|
| Villanueva CJ | 2020 | Brown fat room temperature and cold | https://www.ncbi.nlm.nih.gov/geo/query/acc.cgi?acc=GSE135391 | NCBI Gene Expression Omnibus, GSE135391 |

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
