## [Decision Letter]

**Acceptance summary:**

This work addresses the important question of what fuels are required to maintain optimal heat production by brown adipose tissue during cold exposure. The study shows that mitochondrial pyruvate uptake is essential for optimal thermogenesis, as conditional deletion of Mpc1 in brown adipocytes leads to impaired cold adaptation. Loss of MPC1 protein in brown adipocytes increased blood levels of 3-hydroxybutyrate levels in cold exposure, suggesting that fatty acids and ketones likely provide an alternative fuel source that may compensate when cytosolic pyruvate is not able to be oxidized. Importantly, the results show that complete glucose oxidation is essential for optimal brown fat thermogenesis.

**Decision letter after peer review:**

Thank you for submitting your article "Mitochondrial pyruvate carrier is required for optimal brown fat thermogenesis" for consideration by *eLife*. Your article has been reviewed by two peer reviewers, and the evaluation has been overseen by a Reviewing Editor and Mark McCarthy as the Senior Editor. The reviewers have opted to remain anonymous.

The reviewers have discussed the reviews with one another and the Reviewing Editor has drafted this decision to help you prepare a revised submission.

There was agreement on the importance of your data showing that cold exposure increased fuel oxidation of glucose and amino acids in brown adipocytes, and the changes were accompanied by enhanced pyruvate uptake and utilization into the brown fat mitochondria. Consistent with these observations, genetic deletion of mitochondrial pyruvate carrier MPC1 in BAT caused cold intolerance in mice. The BAT-MPC1 KO mice exhibited modest but significant glucose intolerance, whereas no difference was seen in insulin sensitivity.

However, there are a number of issues that were deemed important for you to consider in revising the manuscript before final consideration for publication.

1) Can you exclude that there is generalized mitochondrial dysfunction in BAT leading to an inability to defend body temperature. In Figure 6B, KO BAT is depleted in a number of TCA cycle intermediates, which could be due to reduced pyruvate oxidation, but could also be due to general defects in mitochondrial metabolism or anaplerotic depletion. The Materials and methods indicate that labeled palmitate was used in some of the in vitro studies, but these data are not discussed or shown. Was flux of palmitate normal or increased? Do KO adipocytes show that respiration rates were normal with gene deletion rather than the acute inhibitor. Also, pyruvate carboxylation is ignored. This could be important for several reasons. In Figure 5, there is considerable M+3 labeling of malate after CL treatment, which is impaired by MPC1 KO. Could this be due to carboxylase activity? If so, reduced flux through this pathway could also factor into potential TCA cycle impairment or other metabolic fates of pyruvate like lipogenesis.

2) Figure 5 is confusing because significant amounts of labeled TCA intermediates were detected in MPC1 KO cells after CL treatment. Does it indicate that alternative pathways exist to import labeled pyruvate into the mitochondria independent of MPC? On a similar note, the authors need to provide some explanations as to why lactate levels were not different between WT vs. KO after CL treatment. If pyruvate transport into the mitochondria is blocked in the absence of MPC, one would expect changes in labeled lactate or alanine in KO cells.

3) BAT-MPC1 KO mice were hypothermia even though other fuel sources, such as fatty acids and amino acids, were available in the circulation. Also, RER of KO mice was higher than control mice at 6˚C, suggesting that KO mice utilize glucose rather than fatty acids. These data appear inconsistent with the idea that increased ketogenesis compensates for the loss of MPC1 in vivo. Also, In the glucose tracing study (Figure 2), the authors need to discuss changes in TCA intermediates, such as citrate, a-KG, succinate, etc., from glucose after CL^-^316,243 administration. The data would provide important insights into time-dependent changes in glucose utilization in the TCA cycle of brown fat mitochondria.

Summary:

Please carefully consider the above major issues, and revise the manuscript in response to these concerns raised (or respond to the editor if you disagree).

---

## [Author Response]

There was agreement on the importance of your data showing that cold exposure increased fuel oxidation of glucose and amino acids in brown adipocytes, and the changes were accompanied by enhanced pyruvate uptake and utilization into the brown fat mitochondria. Consistent with these observations, genetic deletion of mitochondrial pyruvate carrier MPC1 in BAT caused cold intolerance in mice. The BAT-MPC1 KO mice exhibited modest but significant glucose intolerance, whereas no difference was seen in insulin sensitivity.However, there are a number of issues that were deemed important for you to consider in revising the manuscript before final consideration for publication.1) Can you exclude that there is generalized mitochondrial dysfunction in BAT leading to an inability to defend body temperature. In Figure 6B, KO BAT is depleted in a number of TCA cycle intermediates, which could be due to reduced pyruvate oxidation, but could also be due to general defects in mitochondrial metabolism or anaplerotic depletion. The Materials and methods indicate that labeled palmitate was used in some of the in vitro studies, but these data are not discussed or shown. Was flux of palmitate normal or increased? Do KO adipocytes show that respiration rates were normal with gene deletion rather than the acute inhibitor. Also, pyruvate carboxylation is ignored. This could be important for several reasons. In Figure 5, there is considerable M+3 labeling of malate after CL treatment, which is impaired by MPC1 KO. Could this be due to carboxylase activity? If so, reduced flux through this pathway could also factor into potential TCA cycle impairment or other metabolic fates of pyruvate like lipogenesis.

In order to address the question regarding mitochondrial dysfunction and determining whether there is a respiratory defect with MPC1 deletion, we isolated mitochondria from brown adipose tissue (BAT) of control and Mpc1 knockout mice. Mitochondria were analyzed using the Oroboros O2k system used for high resolution respirometry. Our findings showed that in the presence of pyruvate/malate there was reduced oxidative capacity. However, when mitochondria are incubated with palmitoyl-carnitine, we found a greater increase in oxygen consumption with the loss of MPC1. This would suggest two things, 1) mitochondria from MPC1-deficient cells have compensated by increasing fatty acid oxidation and 2) the mitochondrial respiratory capacity has not generally been impaired at this stage. In addition, when mitochondria are given ascorbate and TMPD, there is equal maximal respiratory capacity. These findings would argue against general mitochondrial impairment.

*“Was flux of palmitate normal or increased?”* We’ve now included the palmitate labeling data. We find that both M+2 malate and M+2 Glutamate were increased in MPC1 null brown adipocytes, to levels that were achieved with CL^-^316,243 administration. These findings point to a compensatory mechanism that leads to enhanced fatty acid oxidation.

2) Figure 5 is confusing because significant amounts of labeled TCA intermediates were detected in MPC1 KO cells after CL treatment. Does it indicate that alternative pathways exist to import labeled pyruvate into the mitochondria independent of MPC? On a similar note, the authors need to provide some explanations as to why lactate levels were not different between WT vs. KO after CL treatment. If pyruvate transport into the mitochondria is blocked in the absence of MPC, one would expect changes in labeled lactate or alanine in KO cells.

We found a significant decrease in labeling of TCA intermediates when MPC1 is deleted (Figure 6B). But as noted by reviewer, there is still some labeling of the TCA pool in MPC1 KO cells.

While deletion of MPC in brown fat prevents pyruvate from directly entering the mitochondria, it does not exclude use of “back door pathways” that may arise to compensate for the loss of mitochondrial pyruvate intake. Others have shown that there are a few compensatory mechanisms, for example transamination of alanine to pyruvate in the mitochondria, glutamine anaplerosis via glutamate to a-ketoglutarate, or conversion of malate that was derived from glutamine to pyruvate through malic enzyme(Bender and Martinou, 2016; Gray et al., 2015; McCommis et al., 2015; Schell et al., 2014; Vacanti et al., 2014; Yang et al., 2014). While under basal conditions (no CL treatment) we see significant increase in labeled lactate in cells lacking MPC1 compared to control cells, it is possible that with the CL^-^316,243 treatment lactate production has already peaked, thus preventing us from seeing further differences in levels of labeled lactate. Alternatively, with CL^-^316,243 treatment we see increased serine labeling, potentially explaining why we do not see further increase in 13C incorporation into the lactate. Another possibility is that lactate produced in the cytosol is converted back to pyruvate in the mitochondrial matrix.

3) BAT-MPC1 KO mice were hypothermia even though other fuel sources, such as fatty acids and amino acids, were available in the circulation. Also, RER of KO mice was higher than control mice at 6˚C, suggesting that KO mice utilize glucose rather than fatty acids. These data appear inconsistent with the idea that increased ketogenesis compensates for the loss of MPC1 in vivo. Also, In the glucose tracing study (Figure 2), the authors need to discuss changes in TCA intermediates, such as citrate, a-KG, succinate, etc., from glucose after CL^-^316,243 administration. The data would provide important insights into time-dependent changes in glucose utilization in the TCA cycle of brown fat mitochondria.

While the respiratory exchange ratio (RER) of 0.7 is indicative of mixed fat use, a ratio of 1.0 suggests the exclusive use of carbohydrates(Deuster and Heled, 2008). When RER values are typically between 0.80 and 0.88, fatty acids are considered the primary fuel. In Figure 4F, *Mpc1*^F/F UCP1-Cre^ mice have RER higher than 1 at the beginning of the experiment which would suggest that these mice may have higher CO_2_ production possibly due to hyperventilation and the increased buffering of blood lactic acid (Ramos-Jimenez et al., 2008). If this is the case, the RER would reflect high lactate levels rather than substrate usage and could potentially explain why these mice have higher RER over period of 4 hours of cold exposure compared to their littermate controls. The TCA intermediates are shown with or without CL^-^316,243 treatment in Figure 6. In control cells, there is significant labeling in response to CL^-^316,243 treatment.